# Dynamic expectations: Behavioral and electrophysiological evidence of sub-second updates in reward predictions

Déborah Marciano [1,2 ✉], Ludovic Bellier [1], Ida Mayer [1,2], Michael Ruvalcaba[1], Sangil Lee [1], Ming Hsu[1,2] & Robert T. Knight [1,3 ✉]

Expectations are often dynamic: sports fans know that expectations are rapidly updated as games unfold. Yet expectations have traditionally been studied as static. Here we present behavioral and electrophysiological evidence of sub-second changes in expectations using slot machines as a case study. In Study 1, we demonstrate that EEG signal before the slot machine stops varies based on proximity to winning. Study 2 introduces a behavioral paradigm to measure dynamic expectations via betting, and shows that expectation trajectories vary as a function of winning proximity. Notably, these expectation trajectories parallel Study 1's EEG activity. Studies 3 (EEG) and 4 (behavioral) replicate these findings in the loss domain. These four studies provide compelling evidence that dynamic sub-second updates in expectations can be behaviorally and electrophysiologically measured. Our research opens promising avenues for understanding the dynamic nature of reward expectations and their impact on cognitive processes.

[1] Helen Wills Neuroscience Institute, University of California, Berkeley, Berkeley, CA, USA. [2] Haas Business School, University of California, Berkeley, Berkeley, CA, USA. [3] Department of Psychology, University of California, Berkeley, Berkeley, CA, USA. ✉email: dvorah.marciano@gmail.com; rtknight@berkeley.edu

On December 18, 2022, 90 min into the final World Cup game with France opposing Argentina, soccer fans all over the world were holding their breath. The game had started with a clear advantage for Argentina. At halftime, Argentina led by two goals. But with less than 10 min left in the game, France brought the score to a tie. During the next 30 min of overtime, Argentina got the lead back, but France once again leveled the score after a late penalty. The game went to a penalty shootout, a best-of-ten tie-breaking method, that finally ended with Argentina winning the cup. For Argentinean and French supporters alike, the game was a roller-coaster ride, with the expectations of seeing their team winning going up and down, and up again.

Reward expectations play a critical role in how people process outcomes and make decisions, and have been extensively researched in psychology, economics and neuroscience[1–8]. Much of this research has examined expectations as static, that is, fixed in time. However, as our soccer example illustrates, we intuitively know that expectations can vary rapidly even in the sub-second time range. Here using a slot machine task that elicits rapid changes in expectation, we provide behavioral and electrophysiological evidence of moment-to-moment changes in expectations.

Reward expectations have a central role in cognition with powerful effects on learning and performance[9–11]. Expectations enhance preparatory attention and reduce stimulus conflict[12], prioritize which information should be stored in working memory[13] and guide cognitive control allocation[14]. Expectations also have a strong effect on affect, as illustrated by the placebo effect[15], or the findings that enjoyment of a film, a vacation[2], a beer[3] or a wine[4] is influenced by expectations about their quality. In risky decision-making, unexpected outcomes were found to have greater emotional impact than expected outcomes[16], and a recent computational analysis showed that happiness ratings in response to a wheel of fortune's outcomes are better predicted by reward expectations and reward prediction errors than by earnings[17]. Expectations can also have substantial effects on choice behavior, for example by encouraging individuals to participate in a lottery[18] or to engage in exploration vs. exploitation[19].

Expectations have traditionally been studied as static predictions about future outcomes. Reward predictions are typically elicited by a single predictive cue, such as a sound, an odor[20], or an image[21,22]. In risky decision-making research, studies have used an array of methods to convey reward probability in order to manipulate expectations, such as varying the position of a horizontal line (high, middle, or low probability of winning)[23], the area of a wheel of fortune associated with a gain[16,17], or the colors of cues[24]. In all these studies, the assumption is that expectations stay constant, with little attention paid to the dynamic evolution of expectations leading up to an outcome. However, as our soccer example illustrates, expectations are often dynamic, especially in situations where new information about the odds of receiving a reward is provided.

Investigating dynamic expectations requires finding an appropriate task and the right methodology. Here we use slot machines as a case study to assess moment-to-moment changes in expectations. In the simplest casino slot machines, players start games either by pushing a button or pulling the handle. The reels spin, then decelerate to a stop, and players are rewarded if matching symbols align on the payline (Fig. 1a). The continuous deceleration phase provides the perfect naturalistic setting to study the formation and changes of rapid reward expectations associated with different outcomes, as symbols get closer to or pass the payline.

Slot machines have been extensively used and validated in the gambling literature[25,26]. To increase their ecological validity, some of these studies have used computerized slot machine paradigms that mimic casino settings, with 3D graphics, sounds and realistic spinning and deceleration phases[27–29]. Notably, because it was not their focus of interest, these studies did not examine the dynamic aspect of slot machines and mostly ignored the spinning and deceleration phases. The few studies investigating these phases either looked at the overall brain activity during the spinning independently of the final outcome[30], focused on a single moment during the deceleration[31] or used a static, sequential slot machine game with no spinning[32].

In our studies, we used the slot machine paradigm created by Sescousse et al.[29] (Fig. 1a). We hypothesized that during the deceleration phase of a slot machine game, players' expectations fluctuate as they track the position of the winning item relative to the payline, together with the decreasing speed of the machine. We were particularly interested in differences between different types of No-Win outcomes: Near Win Before (when the machine stopped one item before a match, NWB), Near Win After (when the machine stopped one item after a match, NWA), and Full Miss (when the machine stopped at least two items away from a match, FM) (Fig. 1b). We hypothesized that these different No-Wins would be characterized by unique expectation trajectories. More specifically, we predicted that right before the machine stops, expectations for NWB should be higher than expectations for NWA and FM. Figure 1c illustrates the hypothetical expectation trajectories for all four outcomes, and its legend details our predictions.

Measuring dynamic changes in reward expectations at behavioral and neural levels can be challenging, as it requires: 1) a methodology with a sub-second temporal resolution and 2) a method that does not rely on self-report, as participants might not be able to accurately describe their sub-second internal beliefs. In addition, repeatedly asking participants to report their expectations during a slot machine game would interrupt the experience flow. Here we sought to overcome these challenges using a combination of electroencephalography (EEG) and behavioral methods to investigate the dynamics of expectations.

EEG has an excellent temporal resolution, detecting millisecond changes in brain activity, and it does not rely on self-report nor does it require overt responses. Past EEG studies looked at Near Wins in slot machine games, however they mostly focused on the outcome evaluation phase (once the machine stops) and did not examine the period preceding feedback[28,33–35]. The exception is a study by Alicart et al.[31]. Although the authors emphasize that the stronger effects were observed post outcome, they report that one second before the machine stopped, activity in theta and alpha frequency-bands was higher for Near Wins than Full Misses. However, the paper did not look separately at NWB and NWA, while we predict that these two conditions should show different expectation trajectories. Further, the analyses focus on one timepoint of the deceleration phase (1 s before outcome), and do not examine how the signal evolves during the deceleration phase as expectations are continuously updated. Here we focused on the continuous EEG activity during the deceleration phase. While there is no known EEG metric of dynamic expectations, the prefrontal-dependent contingent negative variation (CNV) event-related potential (ERP) is well known to be linked to static expectations[36–38]. It is characterized by a fronto-central scalp distribution and is maximal at the vertex (electrode Cz)[33,34]. We predicted that CNV-like EEG activity changes would distinguish between the different outcomes.

In parallel, we designed a paradigm, "Slot or Not", to behaviorally measure moment-to-moment changes in expectations and statistically relate them to those observed in EEG responses. This task was inspired by live betting (also known as "in-play betting") which refers to gambling that occurs after a sport or gaming event has started. It offers gamblers the opportunity to identify and capitalize on changing odds during the course of the game. Here we implemented live betting into a slot machine game

## a. Paradigm

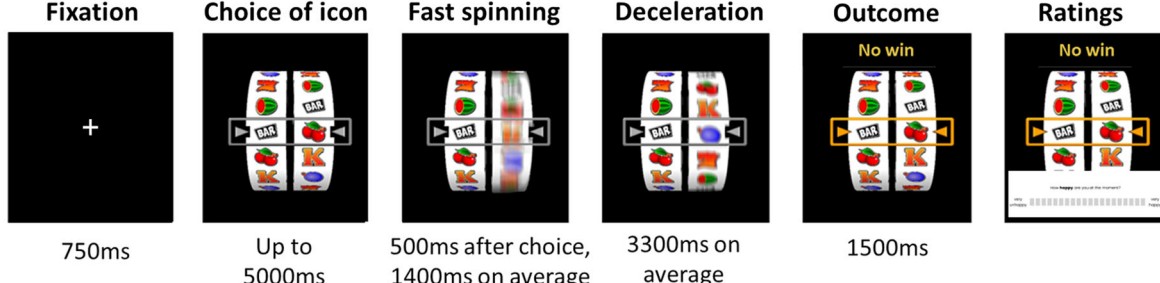

## b. Outcomes

## c. Hypothetical expectations trajectories

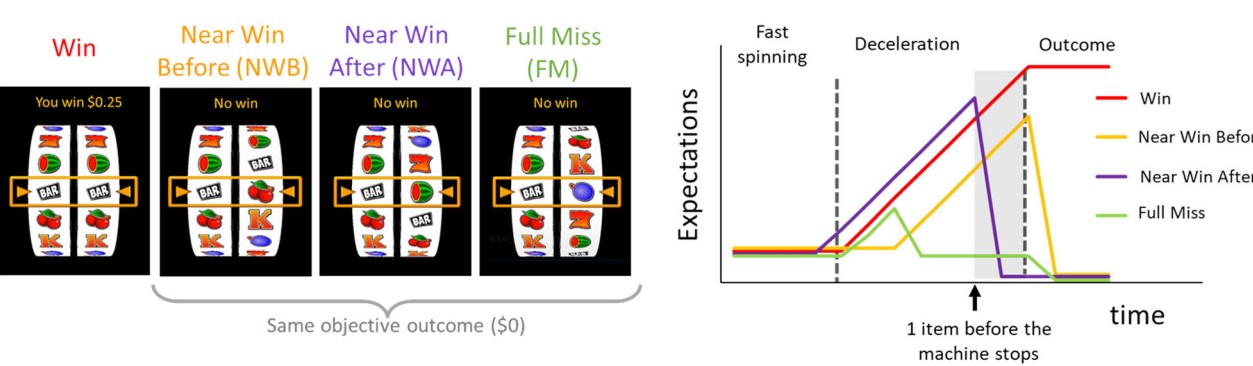

**Fig. 1 Study 1's experimental paradigm and hypotheses. a** *Experimental paradigm*: Each trial started with a 750 ms fixation cross. In the choice phase, participants had a maximum of 5 s to select one of six items on the left reel of the slot machine. They could move the reel downward and upward using the left and right arrow keys on the keyboard, respectively, and validate their choice with the down arrow key. The right reel started spinning 500 ms after choice validation. The fast-spinning phase lasted 1400 ms on average and was followed by the deceleration phase (3300 ms on average), until the machine stopped, marking the beginning of the outcome phase. The outcome phase was characterized by the payline turning orange, and different sounds and written feedback depending on outcome. If the right reel stopped on the same item as the one selected on the left reel (match), participants heard a cash register sound, and in parallel the words "You win $0.25" appeared on the screen. If the right reel stopped on a different item than the one selected (mismatch), participants heard a buzzer sound, and the words "No win" appeared on the screen. Once every five trials on average, participants were asked to rate their happiness and their motivation to play again on a continuous scale ranging from "Not at all" to "Very much". They moved the cursor (always initially positioned in the middle) using the right and left arrow keys, and validated their choice using the down arrow key. **b** *Outcomes*: We classify outcomes depending on the distance from a match. Wins are characterized by a match. In Near Wins Before (NWB), the right reel stopped one item before a match. In Near Wins After (NWA), it stopped one item after a match. In Full Misses (FM), it stopped 2 or 3 items away from a match. Note that NWB, NWA and FM all lead to the same objective outcome ($0), and that they are not signaled differently to participants in terms of sound (buzzer) or written feedback ("No win"). **c** *Hypothetical expectation trajectories for the four outcomes:* X axis represents time and Y axis represents subjective expectations. We hypothesized that during the deceleration phase of a slot machine game, participants track the position of the winning item relative to the payline, resulting in different expectations for Wins (in red), NWB (orange), NWA (purple) and FM (green). We predicted that expectations for FM should decrease earlier, as participants realize the machine is going to stop soon and the winning item is far from the payline. For Wins, NWB, NWA, we predicted an increase in expectations during the deceleration phase, as the winning item gets closer to the payline. Our main prediction regards the last moments before the machine stops, and precisely the time window between the passage of the second to last item on the payline, and the machine coming to a standstill (highlighted in gray). In the case of a Near Win After, participants see that their chosen item is already on the payline, but that the machine is not yet coming to a stop. As they realize they have lost, their expectations should drop. In the case of Near Win Before, participants see their item getting closer to the payline, and their expectations are high right before the machine stops, just as in the case of a Win. Right before the machine stops, for NWA and FM, the uncertainty about the outcome of the slot machine has been resolved: participants know they will lose, and their expectations are thus low. For NWB and Wins, however, the uncertainty has not been resolved yet, and expectations are high.

to track expectations via betting behavior. We are not aware of other behavioral tasks investigating the sub-second dynamics of expectations.

In this paper, we present four studies investigating the dynamics of expectations. In Study 1, we used EEG to define the sub-second electrophysiological correlates of moment-to-moment changes in expectations elicited by the deceleration phase of a slot machine. In Study 2, we introduced a paradigm to measure moment-to-moment changes in expectations from behavior. We examined the relationship between the EEG findings of Study 1 and the behavioral findings of Study 2. Finally, in Studies 3 (EEG) and four (Behavioral), we replicated Studies 1 and 2 in the loss

domain, using a modified version of the slot machine where a match is associated with a loss of money, and mismatches with gains. Our results provide evidence that reward expectations are rapid and dynamic, and that they can be tracked in EEG activity and in choice behavior.

## Results

**Study 1: EEG activity during deceleration varies by outcome consistent with predictions.** We first examined EEG activity during the computerized slot machine game (Fig. 1a) to test the prediction that different types of misses (No-Wins) are associated

with different EEG signals during the deceleration phase. Participants encountered 25 Wins, 25 Near Wins Before (the reel stopped one symbol before the selected symbol), 25 Near Wins after (the reel stopped one symbol after the selected symbol), and 75 Full Misses (the right reel stopped two or three positions away from the selected symbol). The EEG preprocessing steps (details in Methods) followed what has been done in past studies, except for the choice of the baseline correction period. Baseline correction is a common EEG preprocessing step, consisting of subtracting the mean activity of a baseline period from the activity of interest[35]. Notably, previous Near-Wins studies baselined the EEG activity to the period preceding the outcome reveal[28,31,39–41]. However, using the 100 or 200 ms preceding the wheel stop is problematic because this is precisely when expectations might differ the most between the different conditions. Here we baselined EEG activity to the 200 ms period preceding the spinning onset.

To test whether different outcomes elicited different EEG activity during the deceleration phase of the slot machine, we divided the deceleration phase into six 500 ms time-windows, starting 3 s before the machine stopped (the deceleration phase lasted on average 3.3 s). For each of these time-windows we ran a one-way repeated-Measure ANOVA analysis (four outcomes: Win, NWB, NWA, FM), with a significance threshold set at $p = 0.008$ to account for the six time-windows (Bonferroni correction, see Methods). Greenhouse–Geisser correction for analysis of variance ANOVA tests was used whenever appropriate. If the ANOVA was significant, we performed pairwise comparisons using Tukey tests. Here we report the findings of interest. Full results are presented in the Supplementary materials (Supplementary Tables 1–6).

At deceleration onset ([−3000 −2500] and [−2500 −2000] time windows) there was no effect of Outcome (all $p$'s > 0.15). As the deceleration progressed, the effect of Outcome emerged in the [−2000 −1500] time-window ($F(3, 105) = 5.72$, $p = 0.003$). This effect increased in the [−1500 −1000] time-window ($F(3, 105) = 14.87$, $p = 0.001$). Here we detail these effects for the last second before the machine stopped.

In the [−1000 −500 ms] time window, pre-wheel stop EEG activity differed depending on Outcome ($F(3, 105) = 27.50$, $p < 0.001$). Pairwise comparisons revealed that EEG amplitude was smaller for Win ($-7.26 \pm 6.57\,\mu$V) compared to NWB ($-2.52 \pm 5.13\,\mu$V; $t = -5.95$, $p < 0.001$) and Full Miss ($-1.85 \pm 3.31\,\mu$V; $t = -6.78$, $p < 0.001$), but not compared to NWA ($-7.31 \pm 5.25\,\mu$V; $t = 0.06$, $p = 1$). NWA were also smaller than NWB ($t = -6.01$, $p < 0.001$) and FM ($t = -6.84$, $p < 0.001$). There was no significant difference between FM and NWB ($t = 0.83$, $p = 0.84$). As can be seen on Fig. 2a, we observed two pairs of outcomes: Wins and NWA with an enhanced CNV-like negativity, and NWB and FM with no pre -wheel stop negative shift. Expectations are lower for the latter pair: for FM, participants realized early during the deceleration that they were not likely to win ("my item just passed the payline and the machine is slowing down, it won't have the energy to complete a full spin"), and for NWB, the selected item is still farther away from the payline. (Note that all the data used to create the figures are provided in Supplementary Data 1).

In the [−500 0 ms] time window EEG activity also differed depending on Outcome ($F(3, 105) = 13.37$, $p < 0.001$). The pairwise comparisons revealed a different pattern of results than in the prior deceleration time window. EEG amplitudes were smaller for Win ($-6.83 \pm 5.67\,\mu$V) than for NWA ($-2.36 \pm 4.41\,\mu$V; $t = -5.50$, $p < 0.001$) and FM ($-3.22 \pm 3.36\,\mu$V; $t = -4.44$, $p < 0.001$), but not compared to NWB ($-5.78 \pm 5.76\,\mu$V; $t = -1.28$, $p = 0.58$). NWB were also smaller than NWA ($t = -4.21$, $p < 0.001$) and FM ($t = -3.15$, $p = 0.011$).

There was no significant difference between FM and NWA ($t = 1.06$, $p = 0.715$). As can be seen in Fig. 2.a, we observed two new pairs of outcomes: Wins and NWB with an enhanced CNV-like negativity, and NWA and FM with a reduced negative shift. This result is in line with our predictions. Indeed, at around −685 ms on average, the one-before-last item enters the payline. In the case of a NWA, this is the item participants selected, meaning that at this moment, there is a match on the payline. As soon as they realize the machine is not stopping, attentive participants know they have lost—hence the abrupt change in EEG activity. For NWB, the winning item is getting closer to the payline, and expectations rise—as reflected by the enhanced CNV-like EEG negativity. Topographies for the different windows of interest are shown in Fig. 2b.

In addition, in line with the EEG literature on slot machines[28,31,39–41], we examined two ERPs elicited by the outcome phase: the Feedback Related Negativity (FRN) and the P3 (see Methods). The full results are presented in Supplementary Tables 7, 8 and in Supplementary Fig. 1. In line with past studies, we found that the FRN and the P3 were larger for Wins than No-Wins (all $p$'s <, $p < 0.001$)[42–44,45–48]. Notably, NWB elicited larger P3 than the other No-Wins (all $p$'s ≤ 0.001). Given that P3 has been associated with surprise and reward prediction errors[48–50], these results reinforce the claim that NWB create higher expectations than NWA and FM right before outcome onset.

Study 1 assessed sub-second changes in expectations: we found evidence that EEG activity tracked expectations while the slot machine decelerated and participants accumulated information. In Study 2, we measured moment-to-moment changes in expectations via behavior.

## Study 2: Behavioral evidence of moment-to-moment changes in expectations and association of behavioral results with EEG responses. 
In Study 2, we implemented a paradigm to measure dynamic expectations via behavior, incorporating live betting into a slot machine game. In our "Slot of Not" paradigm (Fig. 3a), participants were presented with two options: a sure amount of money, and a slot machine associated with a potential high gain (this machine was the same one as the one used in Study 1, see Methods). Once participants chose one option, the slot machine started spinning, and then decelerated to a stop. The critical part of the game is that participants were allowed to switch between options as often as they wanted while the machine spun. Participants were incentivized to report their true expectations at each timepoint: they were informed that their final payment would depend on their choice at a random timepoint t (Fig. 3b). For each trial, we obtained a timeseries of 0 and 1 values (0 = participant chose the sure amount, 1 = they chose the slot machine) from the beginning of the trial to the stop of the machine. Figure 3c show the expectations curves obtained from the aggregating these timeseries across trials and participants.

We used the same approach as for the EEG data: we divided the deceleration phase into six 500 ms time-windows, and ran a one-way repeated-Measure ANOVA analysis (four outcomes: Win, NWB, NWA, FM) for each time-window. The significance threshold was set at $p = 0.008$ to account for the multiple time windows (Bonferroni correction). Greenhouse–Geisser correction for analysis of variance ANOVA tests was used whenever appropriate. If the ANOVA was significant, we performed pairwise comparisons using Tukey tests. Full results are presented in Supplementary Tables 9–14.

We found no main effect of Outcome for the [−3000 −2500], [−2500 −2000] and [−2000 −1500] time windows (all $p$'s > 0.2).

A significant effect of Outcome was found for the [−1500 −1000] time window ($F(3,87) = 11.34$, $p < 0.001$). In that time

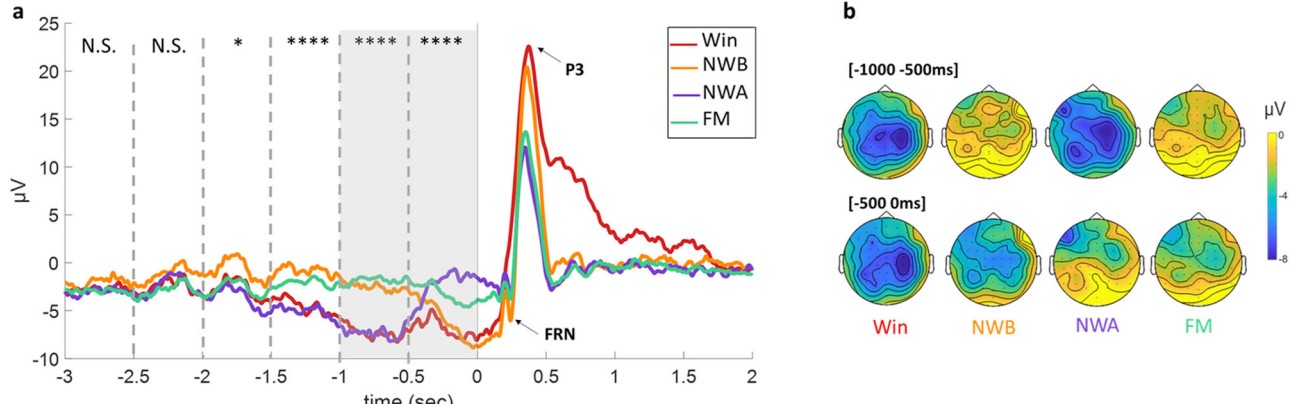

**Fig. 2 Study 1's EEG results. a** EEG activity at Cz during the deceleration phase: Grand average ERPs for each outcome locked to the stop of the machine at electrode Cz. The dashed lines indicate the 500 ms time-windows used for the repeated-Measure ANOVA. N.S: ANOVA was Not Significant. *: p-value < 0.01. ***: p-value < 0.001, n = 36 participants. The gray rectangle indicates the last second of deceleration with detailed results provided in the text. The Feedback Related Negativity (FRN) and P3 are shown on the grand averages. See Methods for analysis of the FRN and P3, and Supplementary Fig. 1 for results. **b** Topographies during the last second of deceleration in μV, for each outcome (upper topographies: [−1000 −500 ms], lower topographies: [−500 0 ms]). Topographies of the differences between outcomes can be found in Supplementary Fig. 3.

window, NWA were associated with a higher tendency to bet on the slot machine (0.56, ±0.29) than Wins (0.41, ±0.26, p = 0.002), NWB (0.35, ±0.25, p < 0.001), or FM (0.39, ±0.24, p < 0.001). All other comparisons were not significant.

A significant effect of Outcome was found for the [−1000 −500] time window (F(3,87) = 33.63, p < 0.001). In that time window, NWA were still associated with a higher tendency to bet on the slot machine (0.58, ±0.25) than Wins (0.48, ±0.23, p = 0.023), NWB (0.31, ±0.24, p < 0.001), or FM (0.30, ±0.20, p < 0.001). In addition, Wins were associated with a higher tendency to bet on the slot machine compared to NWB (p < 0.001) and FM (p < 0.001). There was no difference between NWB and FM (p = 0.987).

A significant effect of Outcome was found for the [−500 0] time window (F(3,87) = 19.06, p < 0.001). In that time window, Wins were associated with a higher tendency to bet on the slot machine (0.58, ±0.25) than NWB (0.35, ±0.26, p < 0.001), NWA (0.38, ±0.25, p < 0.001), and FM (0.22, ±0.17, p < 0.001). In addition, FM were associated with a lower tendency to bet on the slot machine compared to NWB (p = 0.043) and NWA (p = 0.009). There was no difference between NWB and NWA (p = 0.941). However, visual inspection of the grand averages (Fig. 3c) reveals that NWB and NWA had opposite trajectories in the [−500 0] time window, with NWA starting higher but decreasing, and NWB starting lower and increasing, potentially masking differences between the two conditions. To test for differences between NWB and NWA, we conducted a post-hoc slope analysis. For each participant and each outcome, in the [−500 0] time window, we calculated the slope coefficient of the expectation trajectory by regressing the tendency to bet over time. We then conducted a paired t-test on these coefficients for NWB and NWA and found that the slopes for these two outcomes differed (t(29) = 5.53, p < 0.001).

Taken together, Study 2's findings confirm that participants' expectations of the slot machine ending with a match varied by outcome during the deceleration phase. The expectation trajectories were similar to our predictions (Fig. 1c), with FM decreasing early in the deceleration phase ("there is no way the machine is going to spin all the way to my item"), NWA peaking earlier than Win and NWB, and then abruptly decreasing ("Oh no, my item is on the payline, but the machine is not stopping, I've lost!"), and Wins and NWB going up until the machine stops ("my item is getting closer and closer to the payline!").

Next, we examined whether single subjects' EEG data during the deceleration phase correlate with the behavioral expectation trajectories. Our rationale was as follows: although the two studies were run on different participants and the tasks are not identical, they use the same slot machine stimuli. Finding that Study 1's individuals' average EEG activity during the deceleration phase parallels the expectations elicited in Study 2 would considerably strengthen the claim that what is being tracked in the EEG data is indeed expectations.

Here we present the rationale of the statistical method we adopted to examine this correlation. Details can be found in the Methods (See also Fig. 5 in the Methods). Because we are dealing with timeseries, our analysis required us to select a time-window on which to calculate the correlation between EEG and behavioral data. Our selection was data-driven and was done as follows. For each possible time-window (t seconds before outcome), we calculated the correlations between each participant's EEG (averaged per condition) and the corresponding group-level behavioral curves. Note that time-windows were of varying lengths depending on t. We then averaged these correlations across participants, and took this average's absolute value. A permutation test was performed to assess the significance of this averaged correlation. Under the null hypothesis that the relationship between the EEG and the behavioral data is not outcome-specific, we permuted the data by randomly shuffling the labels of the four outcomes attached to the individual subjects' EEG signal. For each permutation we repeated the time-search process, identified the time window maximizing the absolute correlation between the behavioral curves and the EEG activity and noted the value of this correlation. The permutation was performed 10,000 times to create a null distribution against which the actual correlation was compared to. We found that the absolute correlation between the EEG activity and the behavioral curves was maximized for the time window of −0.97–0 s previous to the outcome, with a significant absolute correlation coefficient of 0.44 (p < 0.001) (Fig. 3d). In that time window, the average correlations across participants for each one of the outcomes were: Win: −0.8698, NWB: −0.9620, NWA: −0.9298, FM: 0.7611. We also performed an additional analysis allowing for a lag between the behavioral and the EEG timeseries and obtained similar results (see Supplementary Note 2).

These results provide evidence that the differences in EEG activity before the machine stops are not due to a visual artifact

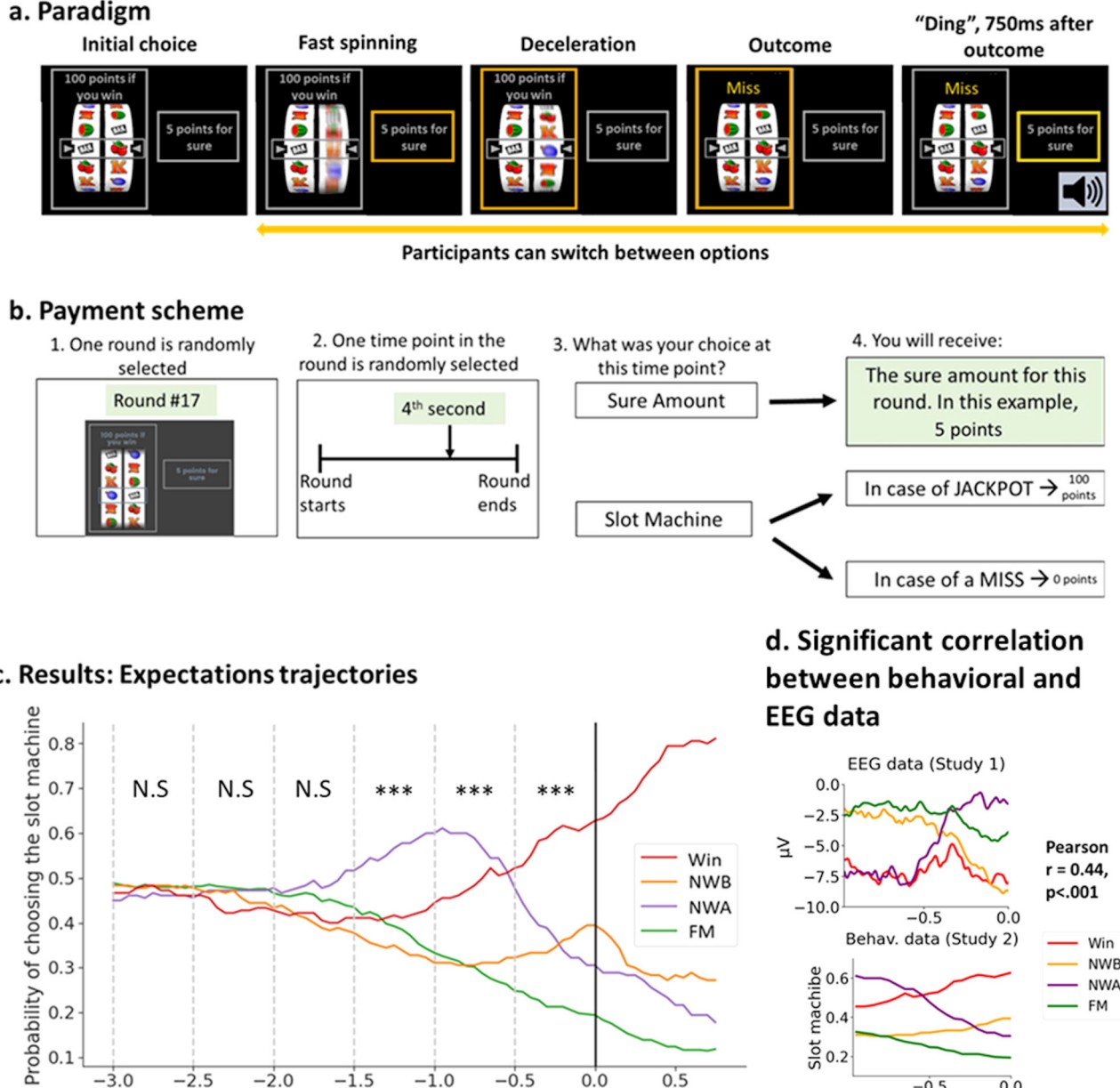

**Fig. 3 Study 2's experimental paradigm "Slot or Not" and results. a** *"Slot or Not" Experimental paradigm:* Participants were presented with two options: a slot machine associated with a potential gain of 100 points, and a sure amount (5 points in this example). Once participants selected one option, the slot machine started spinning and then decelerated to a stop to reveal the slot machine's outcome. If the two items on the payline matched, the word "Jackpot" appeared on top of the slot machine. Otherwise, the word "Miss" appeared. 750 ms after the machine stopped, a "ding" sound marked the end of the trial. Participants could change their choice (switch from the slot machine to the safe amount and vice versa) as often as they wanted during the trial. Their choice appeared as an orange frame around the chosen option. After the "Ding", this frame turned yellow to indicate that their choice was final. **b** *Payment scheme:* Participants were presented with this figure in the instructions in order to clarify the payment scheme. They were told that a trial and a timepoint within that trial would be selected randomly, and that their choice at that timepoint, together with the slot machine's outcome would determine their bonus. The text highlighted in green is only valid for this specific example. If a participant felt the slot machine was about to stop, but the winning item was still far away from the payline, they should have switched as soon as possible to the sure amount option, to maximize the number of timepoints associated with the sure amount. Conversely, if a participant saw that the machine was about to stop and evaluated that the winning item was likely to stop on the payline, they should have promptly selected the slot machine option. Note that while the payment scheme focused on one timepoint only, the analyses was based on all timepoints in the deceleration phase. **c** *Expectation trajectories results:* X axis is time, locked to the machine's stop. Y axis is the probability of choosing the slot machine across participants and trials, for each outcome separately. The dashed lines show the time windows used in the repeated ANOVA analysis. \*\*\* denote time windows in which "Outcome" was significant with a *p*-value < 0.001. N.S: ANOVA was Not Significant. Significance threshold was set at *p* < 008 to account for multiple comparisons, *n* = 30 participants. **d** *Significant correlation between the EEG data (top panel) and the behavioral data (bottom panel).* We found that the absolute correlation between the EEG activity and the behavioral curves was maximized for the time window of −0.97–0 s previous to the outcome, with a significant absolute correlation coefficient of 0.44 (*p* < 0.001). Note that the EEG shown here are grand averages.

from wheel deceleration, but reflect different expectations associated with the four outcomes. There are some notable differences between the results of Studies 1 and 2. For example, In Study 1, we observed a change in the trajectory of NWA around 500 ms before the machine stopped, while in Study 2, the decrease in expectations occurred about 500 ms earlier (1000 ms before the machine stopped). However, this divergence is not surprising given the differences between the tasks and methods used in the two studies. EEG tracks cognitive processes with millisecond precision, whereas behavioral tasks require a decision and a button press, which influences their temporal resolution. Further, in Study 1, participants were obligated to bet on the slot machine, and once they chose an item, they were passive viewers of the reel spinning. In Study 2, participants could make betting decisions during the spinning and deceleration phases, making the game more engaging, and potentially increasing attention to the deceleration phase.

Taken together, Studies 1 and 2 confirm our predictions that Near Win and Full Miss are perceived differently hundreds of milliseconds before the slot machine stops providing evidence that sub-second changes in expectations can be measured and tracked both at the behavioral and electrophysiological levels.

### Study 3: EEG activity during deceleration varies by outcome in the loss domain.

In Study 3, we modified the slot machine game presented in Study 1 so that a match was associated with a "big" loss of money ("Loss", -$.25), and mismatches ("Escapes") with small gains ($.10). This manipulation had two aims. First, it allowed us to test for the robustness of our findings: do they replicate under different task parameters? Second, the comparison between expectations for gains and expectations for losses might provide insights into what precisely is being tracked in the EEG signal.

In this version of the game, participants encountered 25 Losses, 25 Near Loss Before (NLB) (when the machine stops just one item before a loss), 25 Near Loss After (NLA) (when it stops just one item after a loss) and 75 Full Escape (FE) (when it stopped at least two items away from a loss).

Near Losses (also called "Narrow wins") are less studied than Near Wins. They have been implemented in lottery paradigms[51–53] and in the Balloon Analog Risk Task[54], but not in a slot machine game. No study has looked at moment-to-moment changes in expectations in Near Losses using either electrophysiology or behavior. We predicted that right before the machine stops, participants should have smaller expectations of losing (higher expectations of winning) for NLA and FE than for NLB and Loss, for which the uncertainty about the outcome of the slot machine is only resolved when the machine stops.

The analyses presented below followed those performed in Studies 1 and 2. The EEG results during the deceleration phase were similar to those of Study 1 (Fig. 4a). Full results are presented in Supplementary Tables 15–20. At the beginning of deceleration ([−3000 −2500] and [−2500 −2000] time windows), no effect of Outcome was found (all $p$'s > 0.028). However, as the deceleration progressed, differences emerged: the effect of Outcome became significant in the [−2000 −1500] time-window ($F(3, 102) = 7.48, p < 0.001$). This effect increased in the [−1500 −1000] time-window ($F(3, 102) = 9.31, p < 0.001$).

In the [−1000 −500 ms] time window, we found that EEG activity differed depending on Outcome ($F(3, 102) = 17.08, p < 0.001$). Pairwise comparisons revealed that EEG amplitude was smaller for Loss ($-3.85 \pm 5.42 \mu V$) compared to NLB ($-1.25 \pm 4.81 \mu V$; $t = -3.21, p = 0.009$) and FE ($-1.49 \pm 3.74 \mu V$; $t = -2.92, p < .022$), but bigger than NLA ($-6.30 \pm 5.17 \mu V$; $t = 3.03, p = .016$). NLA were also more negative than NLB

($t = -6.24, p < 0.001$) and FE ($t = -5.95, p < 0.001$). There was no significant difference between FE and NLB ($t = 0.30, p = 0.991$). As in EEG study 1, we observed two pairs of outcomes: Loss and NLA (with NLA being more negative than Loss) showing a CNV-like enhanced negativity, and NLB and FE with no negative shift.

In the [−500 0 ms] time window, we again found that EEG activity differed depending on Outcome ($F(3, 102) = 8.83, p < 0.001$). However, the pairwise comparisons revealed a different pattern of results than in the earlier deceleration time window. EEG amplitudes were more negative for Loss ($-5.34 \pm 6.06 \mu V$) than for NLA ($-2.01 \pm 5.18 \mu V$; $t = -4.01, p = 0.001$) and FE ($-2.47 \pm 3.87 \mu V$; $t = -3.46, p = 0.004$), but not compared to NLB ($-5.14 \pm 4.98 \mu V$; $t = -0.24.28, p = 0.99$). NLB were also more negative than NLA ($t = -3.77, p = 0.002$) and FE ($t = -3.22, p = 0.009$). There was no significant difference between FE and NLA ($t = 0.55, p = 0.95$). Again, we observed two new pairs of outcomes: Loss and NLB showed a CNV-like negativity, while NLA and FE did not. Topographies for the different windows of interest are shown in Fig. 4b.

As predicted, we found a larger FRN for Loss than Escapes (all $p$'s < 0.001), confirming that participants understood the meaning of a match in this opposite slot machine game. P3 was also larger for Loss than Escapes (all $p$'s < 0.001). Notably, NLB elicited larger P3 than other Escapes ($p$'s <= 0.001). As for Study 1, these results support the claim that NLB create higher expectations than NLA and FE right before outcome onset. For full FRN and P3 results, see Supplementary Tables 21 and 22 and Supplementary Fig. 2.

### Study 4: Behavioral evidence of moment-to-moment changes in expectations in the loss domain and association of behavioral results with EEG responses.

Finally, in our last study, we modified the "Slot or Not" paradigm to investigate moment-to-moment changes in expectations associated with Losses, NLB, NLA and FE. Figure 4c presents the "grand averages" of the participants' expectations curves for the four outcomes.

As in Study 2, we tested whether different outcomes elicited different expectation curves in the deceleration phase of the slot machine using a repeated-Measure ANOVA. Full results are presented in Supplementary Tables 23–28.

We found no main effect of Outcome for the [−3000 −2500], [−2500 −2000] and [−2000 −1500] time windows (all $p$'s > 0.3). A significant effect of Outcome was found for the [−1500 1000] time window ($F(3,60) = 6.68, p = 0.0023$). In that time window, NLA were associated with a lower tendency to bet on the slot machine (0.53, ±0.31) than Losses (0.74, ±0.26, $p = 0.005$), NLB (0.53, ±0.31, $p = 0.006$), or FE (0.77, ±0.18, $p = 0.001$). All other comparisons were not significant.

A significant effect of Outcome was found for the [−1000 500] time window ($F(3,60) = 10.99, p < 0.001$). In that time window, NLA were still associated with a lower tendency to bet on the slot machine (0.50, ±0.30) compared to NLB (0.75, ±0.26, $p = 0.002$), or FE (0.86, ±0.14, $p < 0.001$). NLA did not differ from Losses (0.60, ±0.31, $p = 0.385$). All other comparisons were not significant.

A significant effect of Outcome was found for the [−500 0] time window ($F(3,87) = 11.23, p < 0.001$). In that time window, Losses were associated with a lower tendency to bet on the slot machine (0.45, ±0.33) than NLB (0.66, ±0.30, $p = 0.026$) and FE (0.86, ±0.15, $p = 0.012$). The comparison to NLA approached significance (0.63, ±0.24, $p = 0.056$). In addition, FE were associated with a higher tendency to bet on the slot machine compared to NLB ($p = 0.027$ and NLA ($p = 0.012$). There was no difference between NLB and NLA ($p = 0.989$). As in Study 1, visual inspection of the trajectories (Fig. 4c) reveals that NLB and

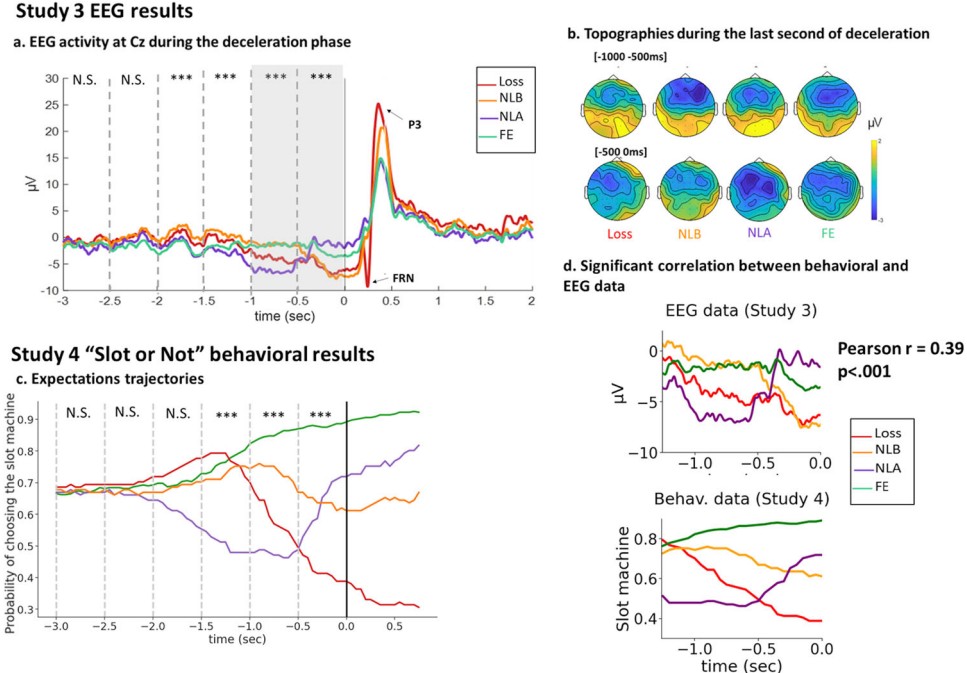

**Fig. 4 EEG and behavioral results for the opposite slot machine (Studies 3 and 4). a** *Study 3 EEG activity at Cz during the deceleration phase:* Grand average ERPs for each outcome locked to the stop of the machine at electrode Cz. The dashed lines indicate the 500 ms time-windows used for the repeated-Measure ANOVA. N.S: ANOVA was Not Significant. *: *p*-value < 0.01. ***: *p*-value < 0.001. Note that threshold for significance was set at *p* < 0.008 to account for multiple comparisons. N = 35 participants. The gray rectangle indicates the last second of deceleration, for which we provide detailed results in the text. The Feedback Related Negativity (FRN) and P3 are shown on the grand averages. See Methods for analysis of the FRN and P3, and Supplementary Fig. 2 for results. **b** *Topographies during the last second of deceleration*, in μV, for each outcome (upper topographies: [−1000 −500 ms], lower topographies: [−500 0 ms]). **c** *"Slot or Not" Expectation trajectories.* X axis is time, locked to the machine's stop. Y axis is the probability of choosing the slot machine across participants and trials, for each outcome separately. The dashed lines show the time windows used in the repeated-measure ANOVA analysis. N.S: ANOVA was Not Significant. Significance threshold was set at *p* < 0.008 to account for multiple comparisons, *n* = 21 participants. **d** *Significant correlation between the EEG data (top panel) and the behavioral data (bottom panel).* We found that the absolute correlation between the EEG activity and the behavioral curves was maximized for the time window of −1.26–0 s previous to the outcome, with a significant absolute correlation coefficient of 0.39 (*p* < 0.001). Note that the EEG shown here are grand averages.

NLA had opposite trajectories in the [−500 0] time window, potentially masking differences between the two conditions. To test for differences between NLB and NLA, we conducted a slope analysis (see Study 1). We found that the slopes for these NLB and NLA differed ($t(20) = -4.04.53$, $p < 0.001$).

Finally, we found that group-level behavioral expectation trajectories correlated with individual subjects' EEG activity prior to the outcome phase observed in Study 3. The absolute correlation was maximized for the time window of −1.26–0 s before the machine stopped, with an absolute correlation coefficient of 0.39 ($p < 0.001$) (Fig. 4d). In that time window, the average correlations across participants for each one of the outcomes were: Loss: 0.2018, NLB:0.5643, NLA:0.7070, FE: −0.6936. We also performed an additional analysis allowing for a lag between the behavioral and the EEG timeseries and obtained similar results (see Supplementary Note 2).

Taken together, Studies 3 and 4 show that Near Losses and FEs are perceived differently hundreds of milliseconds before the slot machine stops. These findings extend Studies 1 and 2 results to the loss domain, and as discussed below, shed light on what cognitive process is being tracked in the EEG signal.

## Discussion
Reward expectations are critical for adjustments in decision-making and reward-seeking behavior. Expectations are likely to evolve in situations in which new information about the odds of receiving a reward becomes available—for example during a horse race or a soccer game. However, little is known about the dynamics of expectations. Here, in four studies (two EEG, two behavioral), we investigated the sub-second dynamics of reward expectations, using slot machines as a test case. In EEG Study 1, we found that different outcomes elicited different EEG activity during the deceleration phase of the machine, indicating that different expectations were formed before the final outcome was revealed. Similar electrophysiological findings were found in EEG Study 3 in the loss domain. In behavioral Studies 2 and 4, we implemented a paradigm designed to track moment-to-moment changes in expectations via betting behavior. We found that different outcomes elicited different expectation trajectories in the deceleration phase. Moreover, we found that these dynamic expectations correlated with the EEG activity in both Study 1 and 3 in the last second of the machine's deceleration. Taken together, these findings provide evidence that reward expectations are rapid and dynamic and can be tracked at the electrophysiological and behavioral levels. Below we discuss these findings, as well as their implications for healthy and unhealthy cognition.

In Studies 1 and 3, we found strong evidence that EEG activity tracks expectations in the slot machine's deceleration phase. In both experiments, the EEG activity over the last second before outcome onset differed for the different outcomes. This result is consistent with the finding that theta and alpha-band activities differ for Near Win vs. Full Miss 1 s before outcome onset[31]. Notably, we found that in that last second, EEG activity changed over time, suggesting that expectations evolved as participants gathered more information about the location of the selected item

on the reel and the speed of the reel. This interpretation is supported by the significant correlation between the EEG activity and the expectation trajectories obtained in the behavioral experiments (Studies 2 and 4). Further, in both experiments, there was no difference between the different outcomes 3 s before reel deceleration (Figs. 2a and 4a) when participants did not have the necessary information to form different expectations.

Insights about what is being reflected in the EEG activity prior to the machine stopping come from the comparison of the findings of Studies 1 and 2 vs. Studies 3 and 4. In Studies 3 and 4, participants played an opposite slot machine game, in which a match was associated with a loss (vs. a gain in Studies 1 and 2). As a result, we predicted that Study 3 and 4's results would be a mirror image of Study 1 and 2's results. This is what we found when comparing the behavioral curves of Studies 2 and 4. However, the similarity between the EEG deceleration findings of Study 1 and Study 3 is striking. As shown in Figs. 2a and 4a, in the last second before the machine stops, the signal for Wins and Losses, Near Win Before and NLB, Near Win After and NLA, and Full Miss and FE are similar. This asymmetry between behavioral and EEG results is best summarized by the sign of the average correlation across outcomes between the EEG and behavioral data. For Studies 1 and 2, this average correlation is negative (Win: −0.8698, NWB: −0.9620, NWA: −0.9298, FM: 0.7611). For Studies 3 and 4, this average correlation is positive (Loss: 0.2018, NLB:0.5643, NLA:0.7070, FE: −0.6936). This suggests that what is being tracked in the EEG during the deceleration phase is not the probability of winning per se (which is high in the case of a Win, but low in the case of a Loss), but rather the certainty of one's outcome, or the certainty of getting a match, no matter the value attached to it. In other words, the EEG activity during the last second of the deceleration phase tracks unsigned expectations. An alternative interpretation is that participants in Study 3 mistakenly believed that a match would yield a gain. However, this interpretation is not likely. While participants might have been initially confused, we believe that the practice trials combined with the buzzer and cash register sound as well as the written feedback helped them overcome the learned default between match and win. In addition, participants' FRN and happiness ratings (see SOM) provide additional evidence that they understood the difference between losses and gains in this unique version of the slot machine game.

The negative shifts observed during the deceleration phase are reminiscent of the well-known contingent negative variation (CNV)[55,56]. The CNV is negative-going ERP deflection traditionally linked to stimulus anticipatory activity. The CNV typically occurs following a stimulus S1 when a motor response to a second stimulus S2 requires maintaining information about S1. In our studies, S1 would be the items passing on the payline during the deceleration, and S2 the machine stopping—which doesn't require any behavioral response. However, the CNV is not a purely motor process: recent studies showed it represents the neural correlate of expectancy for the S2 stimulus[36–38], and is larger for unpredictable targets[36]. This is consistent with the fact that just before the machine stopped, we found enhanced negative amplitudes for uncertain outcomes (Wins and Near Wins Before in Study 1, Losses and Near Losses Before in Study 3) vs. certain outcomes (Near Wins After and Full Misses in Study 1, Near Losses After and FEs in Study 3). In addition, the known fronto-central scalp CNV distribution[33,34] is also consistent with our findings (see topographies in Figs. 2b and 4b). Although there are differences between classical CNV paradigms and our task—especially the dynamic character of our S1 stimulus as opposed to the fixed cue used in the CNV literature - our findings are compatible with a common prefrontal neural sources for both effects[34].

Finally, the P3 findings strengthen the findings that Near Win Before create higher expectations than Near Win After and Full Miss. Indeed, P3 has been associated with surprise and reward prediction errors[48–50]. In Study 1, P3 was larger for NWB compared to NWA and FM, suggesting that participants were more surprised by their loss following NWB. Study 3's results were similar, with larger P3 for NLB vs. NLA and FE. These results are in line with past Near Win studies' findings[28,31,41]. The combined results of Studies 1 and 3 support the finding that P3 tracks unsigned (non-valenced) reward prediction error magnitude[50,57,58]. Further discussion of the P3 and FRN findings can be found in Supplementary Note 3.

One might wonder whether the EEG findings could be accounted by alternative explanations such as basic perceptual features (for example speed of spinning), arousal or attention. Indeed, the deceleration phase is characterized by a change in visual input, as the blurry items presented during the fast-spinning phase become clearer, and stay for longer durations on the payline. Participants might also become more attentive during the deceleration phase, and more aroused, as they know that the trial is about to end. Indeed, cues about the relevance of an upcoming target result in the deployment of selective attention, making it difficult to disentangle the effects of expectations from those of attention[59,60]. However, in our slot machine task, such processes should elicit a similar ramping up (or down) of the EEG signal pre-outcome onset for all four outcomes. Thus, while these processes might account for some of the EEG activity, they cannot explain the unique patterns of activity elicited by the different outcomes. Alternatively, attention/arousal could be correlated with expectations. It could be that during deceleration, participants are more aroused/attentive when the outcome is a Near Win Before vs. a Full Miss because of the uncertainty regarding their outcome, and that this extra arousal/attention is what is being tracked by the EEG signal. This possibility does not undermine or contradict our conclusions that differential sub-second changes in expectations can be tracked at the electrophysiological level.

The "Slot or Not" paradigm used in Studies 2 and 4 was specifically designed to measure sub-second changes in expectations via behavior. Studies 2 and 4's findings confirm our prediction that expectations get updated during the deceleration phase of the slot machine. We developed a statistical method to identify relationships between timeseries and we found that the EEG and the behavioral findings significantly correlate, providing evidence that what is being tracked in the EEG is indeed dynamic expectations.

Reward expectations play a critical role in healthy cognition, and have powerful effects on learning, memory, affect and decision-making. Studies addressing reward expectations typically assume that expectations are static. Our findings confirm what many of us know intuitively: expectations can change from moment to moment. Investigating the temporal dynamics of expectations is crucial if we want to understand how expectations affect us in the real world, when one can accumulate evidence regarding the odds of a certain event and update their expectations, whether it is on the road, at the horse races, or during a romantic date.

Aberrations in how people form expectations play an important role in several mental disorders. In major depressive disorder and anxiety disorders neural responses to anticipated gains are different than those of healthy controls[61–64]. Schizophrenia[65] and attention-deficit/hyperactivity disorder[66] have been associated with reduced striatal BOLD signal during reward anticipation, while pathological gamblers show the opposite pattern[67]. The representation of expected value in the orbitofrontal cortex is also altered in manic patients[68]. Altered expectations likely contribute

to some of the clinical features of these disorders, such as enhanced/decreased motivation for seeking rewards, under/overestimation of risks and maladaptive choice behavior. Examining the temporal dynamics of expectations may shed light on what goes awry in these clinical populations. While slot machine games are of particular relevant to problem gambling, they may be useful for assessing expectation trajectories and reward responsivity in clinical and developmental populations because they are easily understood and easy to play[32]. Our behavioral paradigm, "Slot or Not", is also easy to implement and can be run online. This is a major advantage for studies aiming to recruit participants with disabilities (e.g., patients with Parkinson disease) or with rare disorders.

The Near Miss Effect illustrates how dynamic expectations can add to our understanding of cognition. In the gambling literature, the Near Miss Effect refers to the finding that Near Wins are experienced as less pleasant than Full Misses, yet paradoxically influence a gamblers' actual behavior with bigger money bets following near-misses along with an increased desire to continue gambling[25,41,69]—an effect we also find in Study 1. This effect is so powerful that it is illegal to increase the incidence of Near Misses in Casino slot machines in many jurisdictions[70]. One account of the Near Miss Effect is that Near Wins are mistakenly interpreted as skill acquisition, and thus foster an "illusion of control"[71,72]. Another account of the Near Miss effect is counterfactual thinking: events that almost happened have a stronger emotional impact than events that didn't[41,73,74]. Our findings point to a third, parsimonious explanation: the Near Miss Effect could be the result of the different expectation trajectories leading to the outcome. Expectations and reward prediction errors are a key factor in satisfaction[1,16,17]. Since NWB and NWA have different expectation trajectories toward the end of the slot machine spinning, different effects on affect are predictable. This interpretation is in line with the finding that the Near Miss Effect disappears in paradigms that lack an anticipation phase[52]. Expectations could also explain why Near Miss, like wins, activate regions of the reward network including bilateral ventral striatum and right anterior insula[27,75]. Given that the ventral striatum and insula are also engaged in expectations formation and RPE[76,77], this activity could reflect the high expectations elicited both by wins and Near Win Before right before the machine stops, or the similar RPEs elicited at outcome onset. These events may be masked by the temporal resolution of the BOLD fMRI response. The specific relationship between changes in expectations, motivation and happiness is a subject of future research. Importantly, the illusion of control, counterfactual thinking and expectations accounts of the Near Miss Effect are not mutually exclusive.

To summarize, in a series of 4 studies, we found that expectations are rapidly updated in the deceleration phase of a slot machine game. These findings confirm that expectations are dynamic, and show that sub-second changes in reward expectations can be tracked in EEG activity and in choice behavior. We further examined the relationship between the behavioral and electrophysiological timeseries, and found that the two were significantly correlated. The results open exciting avenues for studying the ongoing dynamics of reward expectations salient to understanding their role in cognition and affect in healthy and clinical populations.

## Methods
We ran four studies: 2 EEG studies (Studies 1 and 3) and two behavioral studies (Studies 2 and 4). In Study 1, we used EEG to define the sub-second electrophysiological correlates of moment-to-moment changes in expectations elicited by the deceleration phase of a slot machine. In Study 2, we introduce a paradigm to measure moment-to-moment changes in expectations from betting behavior. In Studies 3 (EEG) and 4 (behavioral), we replicated Studies 1 and 2 in the loss domain using a modified version of the slot machine where a match was associated with a loss of money, and mismatches with gains. Different participants took part in the different studies.

**Study 1-EEG near wins**. Participants: The experiment was conducted on 42 participants. Data from six participants were excluded because too few artifact-free trials were available. The final sample was composed of 36 participants (21 females, 1 non-binary, mean age = 20.4 years, SD = 1.65, range: 18–24), all undergraduate and graduate students recruited at the University of California, Berkeley. Subjects had normal or corrected-to-normal vision by their self-report, and no history of neurological disorders. Participants were paid $12 per hour to participate in the study. Informed consents were obtained after the experimental procedures were explained. This study, like the three other studies presented in this paper, was approved by the Institutional Review Board at the University of California, Berkeley. In all studies, all ethical regulations were followed.

Stimuli and Procedure: During the task, participants sat in a dark acoustic room and played a computerized slot machine game on a desktop PC. The task lasted for approximately 35 min. It was composed of 150 trials, divided into 3 blocks of 50 trials. Between blocks, participants were given a break and were offered water.

We employed a two-reel slot machine task, identical to the one used in Sescousse et al.[29]. The task was programmed using Neurobehavioral Systems Presentation (version 14.1), incorporating sounds and 3D graphics that made the task realistic and engaging. Participants were informed that their gains in the game were hypothetical. Each trial started with a fixation cross (750 ms) and consisted of four phases: choice, fast spinning, deceleration and outcome (Fig. 1a). In the choice phase, participants selected one of six playing symbols on the left reel of the slot machine. If 5 s passed without a selection, a message appeared on the screen to remind them to make a choice. Following choice, the right reel spun for a variable duration. In the fast-spinning phase (mean = 1430 ms, range: 850–1950 ms), the reel spined at a fast, constant speed. The deceleration phase (mean = 3307 ms, range: 2533–4050 ms) ended with the machine getting to a standstill. In the outcome phase (1.5 s), if the right reel stopped on the same symbol as the one initially selected on the left reel, that is, if they two items on the payline matched, participants heard a cash register sound and saw the words "You win $0.25". In all other cases, participants heard a buzzer sound and saw the words "No win". The sounds had the same sound level. Note that all misses were accompanied by the same sound and visual feedback. Following outcome, on some trials (on average, every 5 trials), participants were asked to answer two questions: "How happy do you feel?" and "How much do you want to play again?" using a continuous scale ranging from "Not at all" to "Very much". These ratings were coded into numeric values ranging from 0 to 10 with an increment of 0.1 (analyses of the happiness and motivation ratings provided by participants at the end of the trials can be found in Supplementary Note 1). Participants completed 3 practice trials before starting the main task. Participants inputted their choices and ratings using the arrows keys on a keyboard. Participants encountered 25 Wins, 25 Near Wins Before (the reel stopped one symbol before the selected symbol), 25 Near Wins After (the reel stopped one symbol after the selected symbol), and 75 Full Misses (the right reel stopped two or three positions away from the selected symbol). The order of trials was randomized across participants.

EEG recording and preprocessing: EEG data was recorded reference-free using an Active 2 system (BioSemi, the Netherlands) with 64 electrodes spread out across the scalp according to the extended 10–20 system (http://www.biosemi.com/pics/cap_64_layout_medium.jpg), and two electrodes placed on the participants' earlobes for offline re-referencing. Horizontal electrooculogram (EOG) were recorded from electrodes placed at the outer canthi of both eyes. Vertical EOG was recorded from an electrode placed below the right eye, and from the right frontopolar electrode FP2. EEG data preprocessing and analyses were conducted using the Fieldtrip toolbox[78] and custom code in MATLAB.

The EEG was continuously sampled at either 1024 or 512 Hz and stored for offline analysis. Offline, EEG data were down-sampled to 512 Hz (when needed), and re-referenced to the average of both ear lobe channels. EEG data were notch filtered at 60 Hz, bandpass filtered between 0.1 and 256 Hz, demeaned and detrended. Electrodes with excessive noise were replaced with an interpolation from neighboring electrodes using spherical spline interpolation[79] via the Fieldtrip function ft_channelrepair. Blink artifacts were removed from the EEG signal using Independent Component Analysis (ICA), combined with manual selection of artifact components based on correlation with the EOG channels and typical component topographies.

Trials were segmented from −3000 ms (beginning of the deceleration phase) to 1000 ms relative to the outcome phase (when the machine stops). Each segment was baseline corrected to the 200 ms preceding the start of the spinning. Trials contaminated by muscle activity, large voltage shifts and amplifier saturation were identified by visual inspection and discarded using the Fieldtrip function ft_reject_visual. Trials were then averaged separately for the four types of outcomes: Win, Near Win Before, Near Win After and Full Miss. These averages were digitally filtered with a low-pass filter at 30 Hz.

**Study 2-behavioral "Slot or Not" near wins**. Stimuli and Procedure: The experiment was conducted online, on the Qualtrics platform, using custom JavaScript scripts. Participants were recruited online through Prolific. They received a $3 participation fee, with an option to win a bonus.

Participants provided informed consent to take part in the study, and then read the tasks instructions. They were tested on their comprehension of the instructions with six questions. If they made any mistake, they were presented with the instructions again. Participants who failed the comprehension questions three times did not get to do the actual task, and received $1 for their time. Following the comprehension test, participants played three rounds of practice and then moved on to the actual task (36 trials, Fig. 3a). Participants who completed the game received a participation fee of $2.75. In addition, they could win a monetary bonus based upon their performances of the game, as detailed below.

Each trial started with two options being presented on the screen: a slot machine and a sure amount. The slot machine was associated with a potential gain of 100 points. The sure amount varied across trials (but never during a trial), and was equal to 1, 5, 10, 15, 20 or 25 points.

Participants selected one option using the right and left arrows on their keyboard. The chosen option was highlighted with an orange frame. The slot machine then started spinning, and then decelerated to a stop. Importantly, participants could change their choice (switch from "slot machine" to "safe amount" and vice versa) as often as they wanted during the trial, up to 750 ms after the machine stopped. To incentivize participants to report their expectations at each given timepoint, we created the following payment scheme. Participants were instructed that one trial, and one timepoint during that trial would be randomly drawn, and that their decision at that timepoint would be implemented. If at that timepoint they chose the sure amount, they would receive the sure amount for this trial as their bonus. If they chose the slot machine, their bonus would depend on the outcome of the slot machine: they would receive 100 points if the slot machine's outcome was a jackpot, 0 otherwise. Participants were told that the chance that their bonus would be determined by the outcome of the Slot Machine (or the Sure Amount) is proportional to the time they spent on that option during the round selected for payment. For example, if they chose the Sure Amount from the beginning and didn't switch, they would always receive this amount. On the other hand, if they spent half on the round on the Slot Machine, there would be a 50% chance that their bonus would be determined by the outcome of the Slot Machine. In other words, we wrote, "each moment you spend on one option increases the chances that that option will determine your bonus". Participants were provided with an illustration of how their bonus would be determined (Fig. 3b). Points were converted to money, with a rate of 1 point = $0.05.

Over the 36 trials of the study, participants encountered 6 jackpots (same items on the payline), 6 Near Win Before, 6 Near Win After, and 18 Full Miss. The order of these trials was randomized for each participant. The position of the slot machine on the screen (right/left) was randomized for each trial.

The slot machine stimuli were the same used in Study 1: we extracted videos from the Presentation experiment and integrated them into the Qualtrics experiment, in order for Study 1 and Study 2 to be comparable. We changed the written feedback that appeared at the outcome phase: instead of "You win $0.25" and "No Win", the words "Jackpot" and "Miss" were displayed.

Participants and exclusion criteria: 51 participants were recruited online. 16 participants failed three times on the comprehension test and did not play the game. We also excluded five participants who did not switch at all in 90% or more of the trials, including in the 750 ms time window following the stop of the machine (when the outcome is already known). Our final sample was thus composed of 30 participants (13 female, mean age = 35.20 year, SD = 11.79, range: 19–64).

**Study 3 – EEG near losses**. Studies 3 (EEG) and 4 (behavioral) aimed to extend the understanding of moment-to-moment changes in expectations to the loss domain. To investigate people's expectations of losing, we modified the slot machine game presented in Study 1. In Study 3, a match was associated with a loss of money, and mismatches with gains. In this mirror-image of a classical slot machine, players could thus encounter Losses, NLB (when the machine stops just one item before a loss), NLA (when it stops just one item after a loss) and FE.

Participants: The experiment was conducted on 41 new participants. Data from 6 participants were excluded because too few artifact-free trials were available. The final sample was thus composed of 35 participants (14 female, mean age = 20.7 years, SD = 2.55, range: 18–28), undergraduate and graduate students recruited at the University of California, Berkeley. Subjects had normal or corrected-to-normal vision by their report, and no history of neurological disorders. Participants were paid $12 per hour to participate in the study. Informed consents were obtained after the experimental procedures were explained.

Stimuli and Procedure: We used the exact same slot machine game as in Study 1. The only difference was the meaning of a match. In this version of the game, if the two items on the payline matched, there was a buzzer sound, and participants lost $0.25. In all other cases, participants won $0.10 and heard a cash register sound. We introduced gains in this paradigm for two reasons: 1) to keep participants motivated during the task, 2) so that the slot machine would have the same expected value as in Study 1. As in Study 1, the monetary gains and losses were hypothetical. Participants encountered 25 Losses (the two items on the payline matched), 25 NLB (the machine stopped one symbol before a match), 25 NLA (the machine stopped one symbol after a match), and 75 FE (the machine stopped two or three positions away from a match).

EEG recording: Recordings settings and preprocessing steps were identical to those of Study 1. As in Study 1, the amplitudes of the FRN and P3 elicited by the outcome phase were measured as their mean value in an 80 ms window around their peak. The FRN elicited by the outcome phase peaked at 248 ms, and we measured it as the mean value in the 210–290 ms window. The P3 peaked at 424 ms, and we measured it as the mean value in the 385–465 ms window. For the deceleration phase, we used the same time windows as in Study 1: −1000 to −500 ms, and −500 ms to 0.

**Study 4–behavioral "Slot or Not" near losses**. Study 4 was a replication of Study 2 in the loss domain.

Stimuli and procedure: We used the exact same paradigm ("Slot or Not") as in Study 2, but changed the meaning of matches. In this version of the game, if the two items on the payline matched, this was a miss, associated with the potential loss of 50 points. If the two items on the payline did not match, this was a jackpot, associated with a potential win of 30 points. Over the 36 trials of the study, participants encountered 6 Losses (same items on the payline), 6 Narrow Escape Before, 6 Narrow Escape After, and 18 FE.

Participants and exclusion criteria: 61 participants were recruited online. 32 participants failed three times on the comprehension test and did not play the game (we suspect some failed on purpose to get the $1 compensation fee). We also excluded 8 participants who did not switch at all in 90% or more of the trials, including in the 750 ms time window following the stop of the machine, that is, when the outcome was already known. Our final sample was thus composed of 21 participants (13 female, mean age = 33.81 year, SD = 9.73, range: 20–58).

**Statistics and reproducibility**. Study 1: For the analysis of the deceleration phase, we used electrode Cz, where the CNV is maximal[55]. We averaged the data into six 500 ms time windows from −3000 ms (beginning of the deceleration) to the stop of the machine ([−3000 −2500 ms], [−2500 −2000], [−2000 −1500], [−1500 −1000], [−1000 −500], [−500 0]). Then, for each time window we ran a one-way repeated measures ANOVA (four outcomes: Win, NWB, NWA, FM). The significance level was set at 0.008 to account for the fact that we tested 6 time-windows, following the formula

$$\alpha' = 1 - (1 - \alpha)^{1/k}, \qquad (1)$$

where $\alpha'$ = Bonferroni correction, $\alpha$ = critical $p$ value (0.05) and k = number of tests (6). Greenhouse–Geisser correction for analysis of variance ANOVA tests was used whenever appropriate. If the ANOVA was significant, we performed pairwise comparisons using Tukey tests with a significance level at 0.05.

Post-outcome, we examined the Feedback Related Negativity (FRN) and P3. The FRN (also known as the Medial Frontal Negativity, MFN) peaks over the scalp's midline within 250–350 ms after feedback is provided, and is generated in part in the anterior cingulate cortex[42,80,81]. The amplitude of the FRN elicited by the outcome phase was measured as its mean value in an 80 ms window centered around its peak. Peaks were determined as follows. Since the FRN has been shown to be maximal at fronto-central midline sites[82,83], we averaged the data across the three frontocentral midline electrodes (Fz, FCz, Cz). The peak of the FRN was then defined as the most negative point in the 200–350 ms time window of the grand average across subjects. The FRN elicited by the outcome phase peaked at 251 ms, and was measured as the mean value in a 210–290 ms window.

The P3 is a large positive component occurring in the 300–600 ms time window after stimulus onset when the stimulus has behavioral consequences. It is generated in frontal and temporo-parietal sites[84] and reflects the summed activity of multiple intracranial sources[85]. Since P3 is maximal at the centro-parietal area[86], we averaged the data across Cz, CPz, Pz, POz, and Oz. The peak of the P3 was then defined as the most positive point in the 300–500 ms time window of the grand average. The P3 elicited by the outcome peaked at 373 ms, and was measured as the mean value in a 330–410 ms window. For both the FRN and the P3, we ran a one-way repeated measures ANOVA (4 outcomes: Win, NWB, NWA, FM). The significance level was set at 0.05. Greenhouse–Geisser correction for analysis of variance ANOVA tests was used whenever appropriate. If the ANOVA was significant, we performed pairwise comparisons using Tukey tests.

Study 2: The behavioral analyses followed the same approach as Study 1's ERPs analyses. For each trial, we obtained a timeseries of 0 and 1 values (0 = participant chose the sure amount, 1 = they chose the slot machine) sampled every 50 ms. We defined six 500ms-time windows of interest from −3000ms (beginning of the deceleration) to the stop of the machine ([−3000 ms −2500], [−2500 −2000], [−2000 −1500], [−1500 −1000], [−1000 −500], [−500 0]). For each subject, in each one of these time windows, we averaged the number of 0 s and 1 s separately for the four outcomes (Win, NWB, NWA and FM). Then, for each time window we ran a one-way repeated-measure ANOVA (four outcomes). The significance level was set at 0.008 (0.05 divided by 6) to account for the fact that we tested six time-windows. Greenhouse–Geisser correction for analysis of variance ANOVA tests was used whenever appropriate. If the ANOVA was significant, we performed pairwise comparisons using Tukey tests with a significance level at 0.05. Figure 3c presents the "grand averages" of the participants' expectation trajectories for the four outcomes.

Do the different outcomes expectations' time course found in Study 2 relate to Study 1's individual subjects EEG activity prior to outcome onset? We will first present an overview of the method we adopted to examine the relationship between the two timeseries; technical details appear below. Figure 5 illustrates parts of the method.

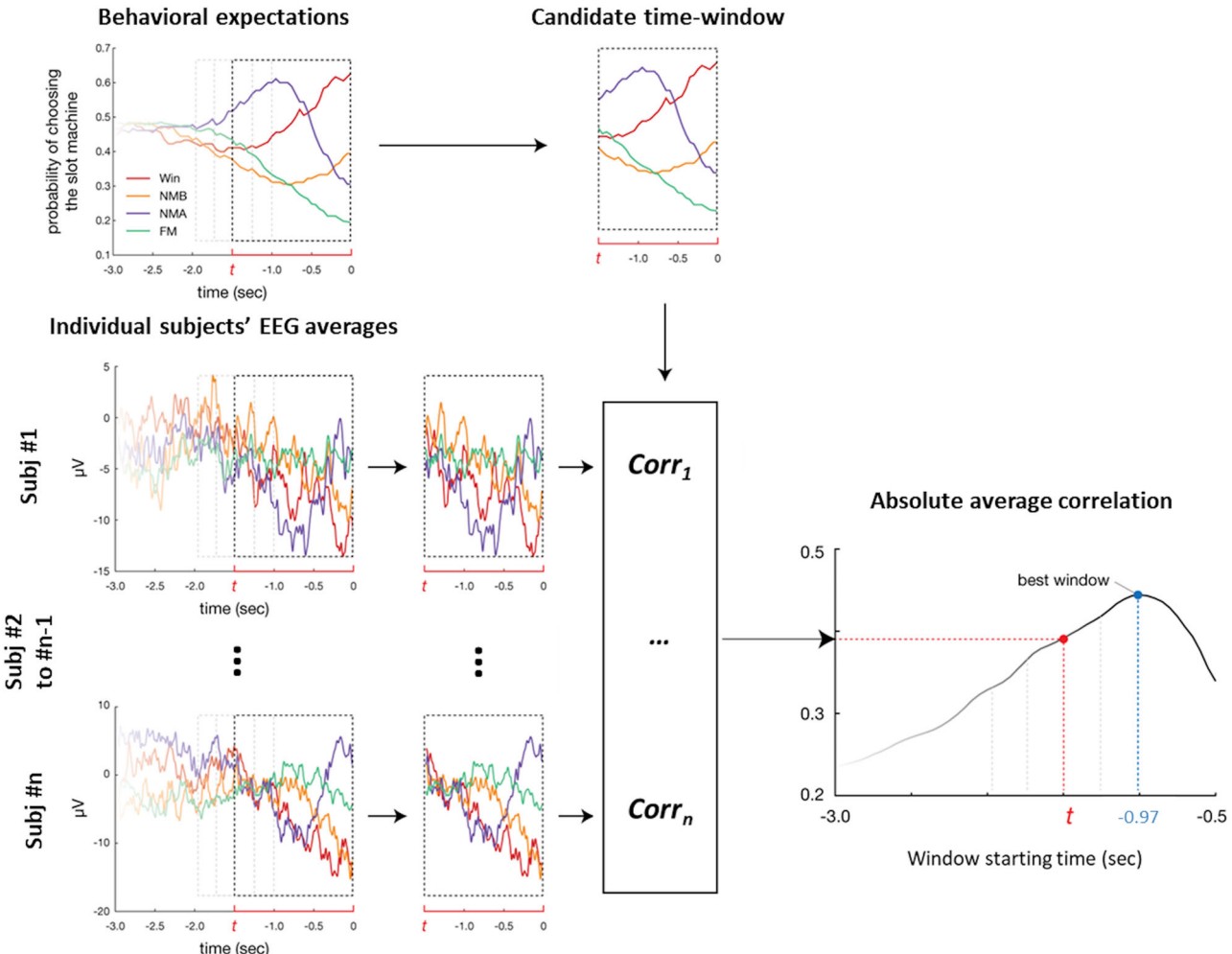

**Fig. 5 Time-search method used for the correlation analysis.** The left upper panel shows the group-level behavioral expectation trajectories from Study 2. The left lower panel shows individual subjects' EEG averages. The dashed rectangles represent candidate time-windows for the correlation analysis. t, in red, represents the window starting point in seconds. Note that depending on t, time-windows varied in length. The upper and lower middle panels show the behavioral and EEG data on a specific candidate time-window. For each possible time-window (starting t seconds before outcome onset), we calculated the Pearson R correlation coefficients between Study 2's group-level expectations curves and Study 1's individual subjects' EEG (See Methods for more details on how this correlation was computed). We then computed the average correlation across subjects, and took the absolute value of that correlation. The lower right panel shows a plot of these absolute average correlations for all the possible time-windows. Finally, we selected the time-window maximizing this absolute average correlation (see Methods). To assess the significance of this average correlation, we then performed a permutation test: we randomly shuffled the labels of the four outcomes attached to the individual subjects' EEG signal (10000 permutations). For each permutation we repeated the time-search process depicted above, identified the time window maximizing the absolute correlation between the behavioral curves and the EEG activity and noted the value of this correlation.

Our analysis required us to select a time window on which to calculate the correlation between EEG and behavioral data. Rather than selecting one based on visual inspection, we performed a data-driven search and looked for the time window (t seconds before outcome) that maximizes the absolute average correlation between the behavioral curves and the EEG activity. For each possible time window (t seconds before outcome), we calculated the correlations between each participant's EEG (averaged per condition) and the corresponding group-level behavioral curves. Note that time-windows were of varying lengths depending on t. We then averaged these correlations across participants, and took this average's absolute value. The time window that maximized this absolute correlation was selected. Second, a permutation test was performed to assess the significance of this correlation. Under the null hypothesis that the relationship between the EEG and the behavioral data is not outcome-specific, we permuted the data by randomly shuffled the labels of the four outcomes attached to the individual subjects' EEG signal. For each permutation, we identified the time window maximizing the absolute correlation between the behavioral curves and the EEG activity and noted the value of this correlation. The permutation was performed 10000 times to create a null distribution against which the actual correlation was compared to.

The parameter search was performed using fmincon in MATLAB using sequential quadratic programming with bounds at −3 and −0.5 s. We set the

minimum duration of time-windows to 500 ms. For each subject, for a given time window -t to 0 s, we interpolated the four behavioral curves and the four EEG curves in 100 timepoints between -t and 0 s to hold the number of data points constant throughout the parameter search process. The correlation coefficient was then obtained by concatenating the four behavioral curves (100 time points each) into one vector and concatenating the four EEG curves (also 100 time points each) into one vector and calculating the Pearson correlation between the two vectors (each with 400 time points).

Study 3: EEG analyses were identical to those performed in Study 1.

Study 4: Analyses were identical to those performed in Study 2.

**Reporting summary**. Further information on research design is available in the Nature Portfolio Reporting Summary linked to this article.

## Data availability

All data reported in this paper were deposited on Zenodo and are available for public download (https://doi.org/10.5281/zenodo.8048351)[87].

## Code availability

Original code was deposited on Zenodo and is available for public download (https://doi.org/10.5281/zenodo.8048382)[88]. Any additional information required to reanalyze the data reported in this paper is available from the lead contact upon request.

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

## Acknowledgements

Deborah Marciano is supported by the Israel Science Foundation fellowship. This work was supported by the following grants: NINDS 2 R01 NS021135 (R.T. Knight) and R01 MH112775 (M. Hsu). We thank Guillaume Sescousse and Luke Clark for sharing their slot machine task's code, as well as Max Good, Jade Green, Amnon Marmor, Kunal Puri and Aastha Shah for their help with data collection.

## Author contributions

D.M. and M.H. designed the experiments; R.T.K. and M.H. supervised data collection; D.M., I.M., and M.R. collected the data; D.M., L.B. and S.L. analyzed the data; D.M., M.H., and R.T.K. interpreted the data; D.M. wrote the manuscript; M.H. and R.T.K. supervised the project.

## Competing interests

The authors declare no competing interests.
