## [Peer Review File · Communications Biology]

Reviewers' comments:

Reviewer #1 (Remarks to the Author):

In this paper, the authors conducted a series of 4 EEG experiments to study neural mechanisms underlying moment-to-moment changes in reward expectations using many variants of slot machine tasks. Across these 4 studies, they have demonstrated converging behavioral and neural evidence showing dynamic sub-second updates in expectations. Moreover, they found the tight relationship between behavioral and electrophysiological timeseries that describe the ongoing dynamics of reward expectations. I would like to applaud the authors' hard work. It is such a great comprehensive work. The study is hypothesis-driven, very-well designed and executed. The report is very well written and easy to follow. The results are robust and replicated within the study. The statistical analyses are well thought-out and performed with appropriate correction methods. The findings are novel and will shed a new light onto our understanding about the neural and behavioral bases of the dynamic processes underlying reward expectation. I think this study will be of the wide interest among people within and across the interdisciplinary fields of brain and behavioral sciences and will open up new approaches for studying reward expectation across healthy and clinical populations.

I only have a few minor suggestions only designed to improve the paper.

In the discussion, the authors expressed concerns about confounding effects of attention/arousal that comes with expectation. Indeed, there is a large body of literature on this issue (e.g., see Kok et al., 2012; Rungratsameetaweemana et al., 2018) worth incorporating into the discussion. I think it would help grab even more audience.

I wonder how the oscillatory data look like. It has been shown that expectation also changes the level of activity of different EEG oscillations including theta, alpha, beta band activity as the authors also mentioned in the introduction (e.g., Alicart et al., 2015; see also Rungratsameetaweemana et al., 2018; Wessel and Aron, 2017). I think looking into this additional information will provide a better mechanistic understanding about the role of neural oscillations in the dynamical processes of reward expectation.

Signed
Sirawaj Itthipuripat

Reviewer #2 (Remarks to the Author):

The authors report a highly innovative and sound series of studies on the temporal dynamics of expectations. Its is methodologically sound and novel. The authors report highly interesting results on the temporal dynamics of expectations in a slot machine task and provide evidence in their electrophysiological and behavioral findings that support their predictions.

The manuscript does not have technical or conceptual flaws. The conclusions are original. The results are of immediate relevance for my discipline. In particular the combination of electroencephalography and of behavioral measures of within trial fluctuations of expectations as realized here are indeed outstanding and novel. I recommend publication after revisions.

Questions:

What is the scale unit for the topographies. Are these simple microvolt topographies or CSD topographies?

Please add whenever appropriate some difference topographies to illustrate the origin of the EEG effect between conditions more precisely.

Analyses on page 11, lines 4 to 20. The between study correlations seem a bit daring and are not fully clear to me yet. Did you consider autoregressive methods that could allow identifying a lag between EEG and behavior? Were the time windows for the correlation always of a specific length? Was the EEG value at each time point correlated with the behavioral indicator at each time point? It was a correlation that assessed the similarity of the slope of the EEG of each person with the slope of the behavioral data? If that is the case the correlations should be reported separately for each of the four behavioral conditions – in my view. That is how similar is the slope of EEG in each condition with the behavioral slope. Figure 3d seems to indicate to me that there are positive (FM) and negative correlations (NWA, NWB) between time series depending upon conditions.

In my view this current correlation analyses “hides” the difference between the win and the loss domain. For example NWA and NLA are very different in their trajectories (Figure 3D and 4D). I would like to see the correlations being calculated for each of the four conditions separately for the two designs (win and loss). This will be relevant for the discussion on page 15/16.

Despite these concerns the study and design is truly outstanding.

Page 3 line 14:

“We hypothesized that these different No-Wins would be characterized by unique 14 expectations trajectories. ”

“We hypothesized that these different No-Wins would be characterized by unique 14 expectation trajectories. ”

Page 7 line 21: we found is twice.

Responses to reviewers

We thank the reviewer for their interest in the paper and their detailed and helpful reviews. We respond to each point below.

Reviewer #1

I only have a few minor suggestions only designed to improve the paper. In the discussion, the authors expressed concerns about confounding effects of attention/arousal that comes with expectation. Indeed, there is a large body of literature on this issue (e.g., see Kok et al., 2012; Rungratsameetaweemana et al., 2018) worth incorporating into the discussion. I think it would help grab even more audience.

Response: We thank the reviewer for pointing to these two pertinent papers. We found the Rungratsameetaweemana et al.'s paper, as well as a number of papers citing it, particularly relevant to our discussion on the potentially confounding effect of attention. We added the following lines into our discussion section:

Discussion (page 17, lines 5-10): "Participants might also become more attentive during the deceleration phase, and more aroused, as they know that the trial is about to end. Indeed, cues about the relevance of an upcoming target result in the deployment of selective attention, making it difficult to disentangle the effects of expectations from those of attention^{59,60}. However, in our slot machine task, such processes should elicit a similar ramping up (or down) of the EEG signal pre-outcome onset for all four outcomes."

I wonder how the oscillatory data look like. It has been shown that expectation also changes the level of activity of different EEG oscillations including theta, alpha, beta band activity as the authors also mentioned in the introduction (e.g., Alicart et al., 2015; see also Rungratsameetaweemana et al., 2018; Wessel and Aron, 2017). I think looking into this additional information will provide a better mechanistic understanding about the role of neural oscillations in the dynamical processes of reward expectation.

Response: We agree with the reviewer that examining oscillatory data will provide an additional layer of understanding in the study of dynamic expectations. Indeed, this is a direction that we are currently pursuing. We have begun this analysis and show below preliminary results on theta band. We feel a full oscillatory analysis is needed to address the role of oscillations in behavior and this work is in progress and the target of a future manuscript. A full analysis will entail in addition to multiple bands network coherence and potential inter-frequency coupling. We provide some theta results below for the reviewer but prefer not to add this to the current ERP paper since it is an incomplete oscillatory analysis.

Based on past papers looking at oscillatory activity in relation with expectations, we examined theta power at electrodes Cz and POz. As in the main ERP analysis presented in the manuscript, we conducted a repeated-measure ANOVA (4 outcomes) on the two time-windows. Greenhouse–Geisser correction was used whenever appropriate. The critical p-value was set at 0.013 after being Bonferroni-corrected for the number of electrodes and time-windows examined. If the ANOVA was significant, we performed pairwise comparisons using Tukey tests.

The figure below presents the theta-activity power as a function of time and outcome in Cz and POz.

At Cz, we found a significant effect of Outcome in the [-500 0] time window ($p < .001$). NWA was significantly higher than Win ($p < .001$), NWB ($p = .03$) and FM ($p = 0.01$).

At POz, we found a significant effect of Outcome in the [-1000 -500] time-window ($p < .001$). Win and NWA were significantly lower than FM ($p < .001$ and $p = .002$). In the [-500 0] time-window, the pvalue ($p = .038$) did not survive the Bonferroni correction. This preliminary analysis is in accord with the papers cited above. In the future we will examine an extended frequency range and multiple electrodes before and after the machine stops, as well as explore the relationships between these different bands.

Reviewer #2

What is the scale unit for the topographies. Are these simple microvolt topographies or CSD topographies? Please add whenever appropriate some difference topographies to illustrate the origin of the EEG effect between conditions more precisely.

Response: The topographies are microvolt topographies. We added the unit to the legend of all relevant figures.

To better illustrate the EEG effect between outcomes, we added to the Supplementary materials the difference topographies between (Win and FM), (NWA and FM), (NWB and FM) and (NWB and NWA), as well as the equivalent for Study 3.

Supplementary Fig. 3 – Difference topographies during the last second of deceleration for Studies 1 and 3

a. Study 1

b. Study 3

Supplementary Figure 3: a. Difference topographies for Study 1 during the last second of deceleration. Upper topographies: [-1000 -500ms], lower topographies: [-500 0ms]). Each topography represents the difference in μV between the topographies for the two outcomes indicated at the bottom. From left to right: Win minus Full Miss, Near Win Before minus Full Miss, Near Win After minus Full Miss, Near Win After minus Near Win Before. **b. Same for Study 3.** From left to right: Loss minus Full Escape, Near Loss Before minus Full Escape, Near Loss After minus Full Escape, Near Loss After minus Near Loss Before.

Analyses on page 11, lines 4 to 20. The between study correlations seem a bit daring and are not fully clear to me yet.

Were the time windows for the correlation always of a specific length?

Response: We thank the reviewer for the opportunity to clarify our approach. The time-windows used to identify the maximal correlation were of varying length. As illustrated in Fig. 5, while all time-windows ended at time = 0 second (when the machine stopped), their starting time point varied, resulting in time-windows of different lengths. We made this point clearer throughout the paper:

Results (page 10, lines 17-19) and Methods (page 23, line 24-27): “For each possible time-window (t seconds before outcome), we calculated the correlations between each participant’s EEG (averaged per condition) and the corresponding group-level behavioral curves. Note that time-windows were of varying lengths depending on t .”

Fig.5's legend (page 24, line 11): "Note that depending on t, time-windows varied in length."

Did you consider autoregressive methods that could allow identifying a lag between EEG and behavior?

Response: We did perform an analysis allowing identification of a lag between EEG and behavioral data by incorporating a lag between the behavioral and EEG data as an additional variable parameter to the model presented in the manuscript (note the original model contained only one variable parameter: the beginning of the time-window). These results are similar to the ones we obtained without the lag. For Studies 1 and 2 (regular slot machine), we found an absolute correlation of 0.45 ($p < .001$) (vs. 0.44, $p < .001$ without a lag) between the two timeseries for a time-window running from -0.97 to 0 seconds for the EEG data and from -0.9 to 0.07 seconds for behavior. For Studies 3 and 4 (opposite slot machine), we found an absolute correlation of 0.39 ($p < .001$) (vs. 0.39, $p < .001$ without a lag) for a time-window running from -1.27 to 0 seconds for the EEG data and from -1.17 to 0.08 seconds for behavior. In both cases, the best correlation was obtained when the behavioral timeseries *preceded* the EEG by 0.07-0.08 seconds.

However, we did not emphasize these results in the manuscript due to ambiguities around interpretation. On one hand, one would expect the EEG to precede behavior, as EEG tracks cognitive processes with millisecond precision, whereas behavioral tasks require a decision and a button press, which introduce a delay. On the other hand, participants did not play the same games in the EEG and behavioral sessions. Participants playing the EEG task may have been less engaged and less attentive than participants in the Slot or Not behavioral task. Indeed, in the EEG task, participants were obligated to bet on the slot machine, and once they chose an item, they were passive viewers of the reel spinning. In the behavioral task, participants could make betting decisions during the spinning and deceleration phases, making the game more engaging, and potentially increasing attention to the deceleration phase.

Following the reviewer's interest, we added the lag results and the explanation above to the Supplementary Materials, as well as a mention of the analysis in the main text:

Results (Study 1, page 10, lines 32-33) and Results (Study 2, page 14, lines 32-33): "We also performed an additional analysis allowing for a lag between the behavioral and the EEG timeseries and obtained similar results (see Supplementary Materials).

Was the EEG value at each time point correlated with the behavioral indicator at each time point? It was a correlation that assessed the similarity of the slope of the EEG of each person with the slope of the behavioral data? If that is the case the correlations should be reported separately for each of the four behavioral conditions – in my view. That is how similar is the slope of EEG in each condition with the behavioral slope. Figure 3d seems to indicate to me that there are positive (FM) and negative correlations (NWA, NWB) between time series depending upon conditions.

In my view this current correlation analyses "hides" the difference between the win and the loss domain. For example NWA and NLA are very different in their trajectories (Figure 3D and 4D). I would like to see the correlations being calculated for each of the four conditions separately for the two designs (win and loss). This will be relevant for the discussion on page 15/16. Despite these concerns the study and design is truly outstanding.

Response: The reviewer brings up an important point we discussed at length in our group when writing the manuscript. We clarify below what we did.

- We examined the correlation between each participant's EEG and the behavioral data at the group level.
- We then examined the correlation across time between the two timeseries. Within a time-window, the values of the EEG and behavioral data at each timepoint constituted a pair of data points. The Pearson correlation measured the strength and direction of the linear relationship between these pairs.
- We did not compute the correlation for each outcome separately. Rather, for each time-window, we concatenated the four EEG curves into a single vector. In other words, we took each outcome EEG trace and pasted to the end of the previous outcome trace, to create one long EEG activity trace instead of 4. We did the same for the behavioral curves while respecting the same order of outcomes.
- To account for the fact that different time-windows would have different numbers of datapoints, we did the following: "For each subject, for a given time window -t to 0 seconds, we interpolated the four behavioral curves and the four EEG curves in 100 timepoints between -t and 0 seconds to hold the number of data points constant throughout the parameter search process. The correlation coefficient was then obtained by concatenating the four behavioral curves (100 time points each) into one vector and concatenating the four EEG curves (also 100 time points each) into one vector and calculating the Pearson correlation between the two vectors (each with 400 time points)." (Methods)

While writing the original manuscript, we did consider the "each outcome separately" approach suggested by the reviewer. However, we eventually decided to opt for the concatenation option for the following reasons.

1. We felt the concatenation approach would be more appropriate to answer the question we were asking.
 - a. We had no specific hypotheses regarding the relative strength of the correlation for the different outcomes. In other words, we wouldn't necessarily know how to interpret a 0.35 correlation for one outcome vs. a 0.72 correlation for some other outcome. Rather, we were interested in understanding if overall, the EEG activity related to subjective expectations.
 - b. The concatenation approach takes into accounts the cross-outcome variances. It assesses the degree to which the EEG signal can predict if an outcome elicits higher expectations than some other outcome. In other words, the concatenation approach also tests whether the order of the outcomes in the EEG signal resembles the order of the behavioral expectations curves.
2. The concatenation approach requires less decisions in terms of analyses/interpretations, and as such is less prone to biases. Testing each outcome separately introduces multiple degrees of freedom in the choice of the analysis parameters. For example: should we look for each outcome separately for the time-window maximizing the correlation between EEG and behavior, or should we select the time-window maximizing the average of the absolute correlations for each outcome? Should the null permutation be calculated for each outcome separately? In terms of interpretation, how would we interpret a situation where three correlations are significant, but not the fourth one?

Nevertheless, based on the reviewer's comment, we calculated the correlation for each one of the four outcomes for the time-window that yielded the highest absolute correlation in the concatenation approach:

Studies 1 and 2: Win: -0.8698, NWB: -0.9620, NWA: -0.9298, FM: 0.7611

Studies 3 and 4: Loss: 0.2018, NLB:0.5643, NLA:0.7070, FE: -0.6936

Two points of interest emerge from these results. The first one is the fact that the correlation for Full Misses (Study 1) and Full Escapes (Study 3) is in the opposite direction as the other correlations in each study. This could be because in the window of interest, the curves for FM and FE are relatively flat, which makes the correlation particularly sensitive to noise. The second and more important one is the fact that while the EEG activity for Win, NWB and NWA is negatively correlated with the behavioral data, the EEG activity for Loss, NLB, and NLA is positively correlated with the behavioral data. The latter point relates, as pointed by the reviewer, to our discussion on pages 15-16 on what is actually being encoded in the EEG signal.

We added the correlation for each outcome in the Results section for each study, as well as a passage in the discussion:

Results (Study 1, page 10, lines 30-32): "In that time window, the average correlations across participants for each one of the outcomes were: Win: -0.8698, NWB: -0.9620, NWA: -0.9298, FM: 0.7611."

Results (Study 2, page 14, lines 30-32): "In that time window, the average correlations across participants for each one of the outcomes were: Loss: 0.2018, NLB:0.5643, NLA:0.7070, FE: -0.6936."

Discussion (pages 15-16, lines 32-9): "Insights about what is being reflected in the EEG activity prior to the machine stopping come from the comparison of the findings of Studies 1 and 2 vs. Studies 3 and 4. In Studies 3 and 4, participants played an opposite slot machine game, in which a match was associated with a loss (vs. a gain in Studies 1 and 2). As a result, we predicted that Study 3 and 4's results would be a mirror image of Study 1 and 2's results. This is what we found when comparing the behavioral curves of Studies 2 and 4. However, the similarity between the EEG deceleration findings of Study 1 and Study 3 is striking. As shown in Figures 2a and 4a, in the last second before the machine stops, the signal for Wins and Losses, Near Win Before and Near Loss Before, Near Win After and Near Loss After, and Full Miss and Full Escape are similar. This asymmetry between behavioral and EEG results is best summarized by the sign of the average correlation across outcomes between the EEG and behavioral data. For Studies 1 and 2, this average correlation is negative (Win: -0.8698, NWB: -0.9620, NWA: -0.9298, FM: 0.7611). For Studies 3 and 4, this average correlation is positive (Loss: 0.2018, NLB:0.5643, NLA:0.7070, FE: -0.6936). This suggests that what is being tracked in the EEG during the deceleration phase is not the probability of winning per se (which is high in the case of a Win, but low in the case of a Loss), but rather the certainty of one's outcome, or the certainty of getting

a match, no matter the value attached to it. In other words, the EEG activity during the last second of the deceleration phase tracks unsigned expectations.”

“We hypothesized that these different No-Wins would be characterized by unique 14 expectations trajectories.”

“We hypothesized that these different No-Wins would be characterized by unique 14 expectation trajectories.”: We thank the reviewer for catching this mistake. We fixed it throughout the paper.

Page 7 line 21: we found is twice: fixed.

REVIEWERS' COMMENTS:

Reviewer #1 (Remarks to the Author):

This is an exceptional paper. I'm fully satisfied with the authors' responses. I'm looking forward to seeing it in publication form and also its future extension examining at oscillatory events.

Signed Sirawaj Itthipuripat

Reviewer #2 (Remarks to the Author):

My comments were all answered excellently. I suggest acceptance. Thank you.